# A Practical Multi-Stage Grasp Detection Method for Kinova Robot in Stacked Environments

**DOI:** 10.3390/mi14010117

**Published:** 2022-12-31

**Authors:** Xuefeng Dong, Yang Jiang, Fengyu Zhao, Jingtao Xia

**Affiliations:** Faculty of Robot Science and Engineering, Northeastern University, Shenyang 110169, China

**Keywords:** grasp detection, multi-stage network, multi-task, stack scenarios, VMRD, Kinova robot

## Abstract

Grasp detection takes on a critical significance for the robot. However, detecting object positions and corresponding grasp positions in a stacked environment can be quite difficult for a robot. Based on this practical problem, in order to achieve more accurate object position detection and grasp position detection, a new method called MMD (Multi-stage network for multi-object grasp detection algorithm) is proposed in this paper. MMD covers two parts, including the feature extractor and the multi-stage object predictor. The feature extractor refers to a deep convolutional neural network that can generate shared feature layers as well as the initial ROIs (region of interest). A multi-stage refiner serves as the multi-stage object predictor, which continuously regresses the initial ROI to obtain more accurate object detection and grasping detection results. Ablation experiments show that the proposed MMD has better grasp detection performance. The specific performance is that the recognition precision achieves a state-of-the-art 76.71% mAPg on the VMRD dataset. Moreover, test experiments demonstrate the feasibility of our method on the Kinova robot.

## 1. Introduction

Grasping is recognized as a basic function for robots, whereas it is quite difficult actually. Existing research [1,2,3,4,5,6] has focused on grasp detection in single object scenes. However, multi-object scenes [7] are common in practical applications. For a graping task on robots, the position of the object and where the object is grasped are both critical. This knowledge not only allows the robot to understand the types of objects to be grasped, but also to accurately find the grasping position of the object. Thus, the main research idea of this study is to use a multi-stage deep neural network to solve the multi-object grasping problem in stacked scenes. At the same time, it also has very good object detection performance. Therefore, our method has good practical value.

In general, deep learning based object detection schemes are divided into one-stage algorithms and two-stage algorithms, which currently include convolutional neural networks (R-CNN [6]), Fast R-CNN [8] and Faster R-CNN [9]. The initial R-CNN is adopted to detect the class and position of objects in the image. R-CNN applys the selective search method to generate regions of proposal, but it also limits the detection speed. In addition, it follows the method of convolution for the respective proposal of each image, thus significantly increasing the computational effort. Fast R-CNN replaces the method of extracting targets with network training, and the detection speed are significantly improved over before. Fast R-CNN shares convolution features with detection networks, which can greatly reduce the computation time of candidate regions. However, the selective search method is still retained. Faster R-CNN adopts the RPN (Regional Proposal Network) to achieve regional proposal generation, thus notably increasing the speed and quality of proposal box generation and addressing the performance bottleneck problem of Fast R-CNN.

Despite the good effect achieved by Faster-R-CNN, while the IOU (Intersection over Union) threshold is selected high, it will cause model overfitting. Besides, a single threshold will cause mismatch problems. A cascade type of network namely Cascade R-CNN [10] solves the mismatch problem from Faster R-CNN. Cascade R-CNN was implemented based on Faster-R-CNN. Cascade R-CNN is a multi-stage method that generates region proposals from the previous stage and feeds them into the next. It applys a shared feature layer to all stages to avoid a tremendous computational effort. Accordingly, Cascade R-CNN comprises two parts, including a feature extractor and multi-stage refiner. A feature extractor is adopted to generate shared feature layers, and then a multistage refiner is employed with increasing IOU thresholds to refine the region proposals. However, the Cascade R-CNN is designed for object detection and is not suitable for grasp detection. On that basis, MMD adds a grasp detection branch based on Cascade R-CNN to increase grasping accuracy. Several novel modules are proposed to make the network suitable for the grasping task. The experimental results prove that MMD is very effective on the VMRD dataset. The contributions of this study include:(1)A Cascade R-CNN implementation based on Faster-RCNN for grasp detection is provided. Our model allows for simultaneous grasp detection and target detection.(2)MMD is an improved multi-stage end-to-end grasp detection model that we proposed. This algorithm performs well on the VMRD dataset in stack scenarios and also shows great environmental adaptability on our homemade test sets.(3)A FRM (feature refine model) is proposed in MMD, thus allowing the network to improve the quality of the region proposal feature map. Its effectiveness in facilitating the detection of grasping has been proven through experiments.(4)A box redistribution strategy is proposed in MMD, which avoids filtering the false positive samples to a certain extent and increases the system's fault tolerance for detection. Experimental results also indicate that it can increase the accuracy of grasp detection.(5)In order to test the practicality of our model, we also carried out experiments on our homemade test sets and our Kinova robot.

## 2. Related Works

The methods of earliest grasping detection mainly focus on the case of a single object. Jiang [1] et al. proposed a five-dimensional vector representation method for grasping detection on a 2D image plane and an algorithm for predicting the grasping box of a given object from an image. This method transforms the problem of grasping detection into finding five-dimensional vectors in the image and provides a solution for the application of depth learning in grasping detection. In 2014, Lenz et al. [11] of Cornell University proved that the five-dimensional grasp representation of 2D images can be projected into 3D space. They use first-depth learning in conjunction with robot grasping technology to detect objects on the plane. Chen et al. [4] established a new grasping detection model, which can more fairly evaluate the grasping proposals. This method can improve not only detection accuracy but also generalization. Yokota et al. [12] realized a new type of grasping detection network deployed on the picking robot, which can predict the picking position in the case of a small number of data sets. This method performs well in picking tasks. Although these methods have achieved good results, they are not suitable for stacking and multi object situations.

Research on the grasping of stacked objects has emerged in recent years. Guo et al. [13] propose a model-free shared convolutional network to predict grasping points and object detection boxes. The model is capable of performing object detection and grasp detection simultaneously, and the experimental result indicates that the shared model outperforms the single model, thus revealing that the two tasks are associated with each other to a certain extent. Another common method is the sliding window method for grasping detection. However, this method is time-consuming. Later, Redmon et al. [14] proposed a network named Multi-Grasp that directly regressed the grasping coordinates from images. The model can predict the grasping location of multiple objects in images that do not exist occlusion. In fact, the real-time is complex, and occlusion is inevitable. Zhang et al. [5] studied the problem of grasp detection in stacking scenarios and proposed a novel multi-object robot grasp detection method. The algorithm consists of two parts. One is to detect the position and type of objects in the image; another is to detect the grasping position, for which a fully convolutional neural network is designed and implemented using a directed anchor frame mechanism. Wu et al. [15] proposed a multi-object grasp detection network that can learn object detection and grasp location detection simultaneously using multi-layer features. However, because of the complexity of the application environment, most of the methods do not fully meet the accuracy requirements of practical applications. In this study, a novel grasp model named MMD is proposed, which is capable of achieving a grasping accuracy by multi-stage regression. Compared with the above methods, it not only has higher detection accuracy but can also better meet the actual grasping requirements. Meanwhile, it also demonstrated robust environmental adaptability.

## 3. Proposed Method

In this section, we present the design of MMD. The system overview is first established in Section 3.1. Next, the network architecture is elucidated in Section 3.2. Subsequently, a useful model named FRM is presented in Section 3.3. Afterward, a novel algorithm named (BR) box redistribution is illustrated, which is capable of combining the low and high confidence box to obtain a more accurate regression in Section 3.4. Lastly, we will show the definition of the loss function in Section 3.5.

### 3.1. System Overview

Our architecture comprises two branches: object detection and grasp detection. The object detection branch obtained class and region proposals, and the grasp detection branch is intended to get the grasping position in ROI from object detection. They are not independent, on the contrary, they are combined and connected by sharing features and regions of interest. Here we divide the whole task into three steps, and the paradigm can be summarized as follows:

Step 1: Generate ROIs. Given a single monocular image as input, RPN generated a series of ROIs similar to Faster-RCNN.

Step 2: ROI Refine. A cascade refiner is adopted to generate efficient foreground proposals from ROIs.

Step 3: Grasp detection and object detection. The proposals generated by cascade regression are taken as input, and the grasping detection branch and object detection branch will output the final result.

### 3.2. Network Architecture

The MMD includs a feature extractor, multi-ROI regression and grasp detection, as shown in Figure 1. ResNet101 is a popular choice for a 2D detector for efficiently converting the RGB image into a feature vector. It is adopted as the backbone of our network to perform feature encoding because of its efficiency and accuracy.

**Feature encoding.** The ResNet101 [16] have five residual convolution blocks, which are defined as $\left\{R_{1},R_{2},R_{3},R_{4,\;}R_{5}\right\}$. The network utilizes {$R_{1},R_{2},R_{3},R_{4,\;}$} to gradually convert the raw image $f\in R^{3\times W\times H}$ into feature vector $S\in R^{1024\times \frac{W}{16}\times \frac{H}{16}}$ with 2×, 4×, 8×, 16× down-sampled size. The feature S will be shared in multi-ROI regress stage.

**Multi-ROI regress.** Here, a series of separate subnetworks with raising IOU thresholds {0.5, 0.6, 0.7} is used to refine region proposals similar as [17]. There are 3 refiners {$S_{1}$, $S_{2},\;S_{3}$} in the network of this study. The (j−1) th region proposals $\;B^{j-1}$ from prior refiner will input next refiner. Feature extractor $\varphi_{j}\left(∙\right)$ process features after deformable ROI pooling operation and output the feature $F^{j}$ for jth refiner. Then, with $F^{j}$, a confidence branch $S\left(∙\right)$ and a box regression branch $R\left(∙\right)$ will generate a new object confidence$\;C^{j}$ and box $\;B^{j}$, respectively. The process can be formulated as:(1)$$F^{j}=\varphi_{j}\left(\;B^{j-1}\right),\;C^{j}=S\left(F^{j}\right),\;\;B^{j}=R\left(F^{j}\right)$$ where j=1,2,3. To ensure that the network of this study exhibits prominent performance, different modules are selected for different refiners. At the same time, to solve all the inherent problems of multistage networks, that is, errors tend to propagate downstream. We add $BR_{1}$ module and $BR_{2}$ module after refiner $S_{2}$ and refiner $S_{3}$, respectively. 

Every refiner but refiner $S_{1}$ adopt Fs module to as feature extractor. Before this, it is necessary to process feature from ROIs into tensors of same size. So, the deformable ROI pooling operation is adopted from DcNv. For refiner $S_{2}\;$ and refiner $S_{3}$, the feature extractor function can be formulated as:(2)$$\varphi_{j}\left(∙\right)=FS\left(DROI\left(\;B^{j-1},\;S\;\right)\right)$$ where DROI is deformable ROI pooling operation from DcNv and FS is the Fast Smooth module. FS module is made up of two convolution layers with kernel size 7×7 and 1×1, respectively, the former is to ensure that the height and width of the feature map are equal to one after sampling and the latter is to reduce the model complexity. 

For refiner $S_{1}$, the feature extractor function can be formulated as:(3)$$\varphi_{1}\left(∙\right)=FRM\left(DROI\left(\;B^{0},\;S\right)\right)$$ where FRM is Feature Refine Module. We will intrude the FRM in Section 3.3.

**Object detection and grasp detection.** The object confidence $C^{3}$ and proposals box $\;B^{3}$ from refiner $S_{3}$ is our object detection result that is same as [18]. Since the grasping position also should be obtained, which corresponds to the respective proposal, grasp detection heads are added to the region of interest from refiner $S_{3}$. In the training stage, IOU will be calculated between ROI and ground-truth box. Subsequently, matched ROI that have highest IOU will be remained to as final ROI which will be pass to next refiner. The function extractor function $\delta \left(∙\right)$ can be formulated as:(4)$$\delta \left(∙\right)=R_{5}\left(DROI\left(B\;,\;S\right)\right)$$ where B is fused box parameters after $BR_{2}$. We will introduce box redistribution in Section 3.4.

Every ROI will be divided into M × N grid cells after DROI operation (both M and N are set to 7 in the experiments), and in each grid cell, a set of oriented anchors is used, which is a set of rectangles with oriented angles. Each grasping box is represented by a 5D vector $\left(x,\;y,\;w,\;h,\;\theta \right)$, as shown in Figure 2. Where the angle θ ranges from 0 to 180 degrees.

A 5 × k offset of the grasping rectangle regression variable in each grid cell relative to the anchor box at the location of the grasping rectangle. The respective 5D vector $\left(\sigma_{x},\sigma_{y},\sigma_{w},\sigma_{h},\sigma_{\theta }\right)$ is computed with respect to the anchor ($p_{x},p_{y},p_{w},p_{h},p_{\theta }$) and the predicted grasping rectangle (x,y,w,h,θ) using Equations (5)–(9).
(5)$$\sigma_{x}=\left(x-p_{x}\right)/\;p_{w}$$
(6)$$\sigma_{y}=\left(y-p_{y}\right)/\;p_{h}$$
(7)$$\sigma_{w}=log\left(w/\;p_{w}\right)$$
(8)$$\sigma_{h}=log\left(h/\;p_{h}\right)$$
(9)$$\sigma_{\theta }=\left(\theta -p_{\theta }\right)/\left(90/k\right)$$ where k is equal to 7. The grasping classifier provides 2×k confidence scores, thus indicating the probability of k anchor frames that are graspable and ungraspable, respectively. Finaly, the grasping detector will predict M×N×k grasping proposals. And the appropriate ROI and the best grasp proposals frames are retained using the non-maximum suppression algorithm (NMS).

### 3.3. Feature Refine Module (FRM)

A feature refine module is introduced in this Section, as illustrated in Figure 3. The pooled features $P_{1}=DROI\left(\;B^{0},\;S\right)$ in the refiner $S_{1}$ will be input to FRM first, and feature extraction can be performed to allow the network to focus on the superior region proposal features. 

The FRM module comprises three branches {$b_{1},b_{2},b_{3}$}, and the convolution layers in these branches all use the Mish activation function. For different branchs, we use different scheme to extract feature. For branch $b_{1}$, its formula can be presented as:(10)$$a=sig\left(AO\left(P_{1}\right)\right)$$ where AO is composed of a 1 × 1 convolution and a 7 × 7 Depthwise Separable Convolution. And sig() is Sigmoid activation function. AO ensures the output features involve spatial features with a larger receptive field, and all values are normalized from 0 to 1.

The second branch is based on the channel attention model. Firstly, the globally pooling operation reduce the input features $P_{1}$’s height and width to 1. Subsequently, two 1 × 1 convolutions are to adjust the channel of feature. Lastly, we adopt the sigmoid activation function to normalize the value to $\left(0,\;1\right)$. Its specific formula can be written as:(11)$$b=sig\{C_{1\times 1}\left(\left(GP\left(P_{1}\right)\right)\right)\}$$ where GP denotes globally pooling; $C_{1\times 1}\;$represents 1 × 1 convolutions.

Afterward, the final feature is obtained as:(12)$$F^{1}=a∙b+P_{1}∙a∙b$$

The final feature map covers both channel attention and spatial attention, thus allowing the network to improve the quality of the region proposal feature map.

### 3.4. Box Redistribution(BR)

In a multi-stage network, the errors tend to propagate along with downstream [17]. To further address this problem, box redistribution is proposed during the testing stage to build more connections between different stages. The idea behind this strategy stems from the fact that each stage produces output with high and low confidence, which can then be combined to produce a more accurate detection box. Notably, a method named “box voting” [17] has been proposed, but it is useless for us. In MMD, features are progressively refined, so the confidence from the refiner $S_{3}$ is considered to be more trustworthy. To solve this problem, a box redistribution (BR) mechanism is adopted, which fuses the box weights by directly weighting the detection box confidence and the detection confidence.
(13)$$C=\frac{1}{N_{r}}\underset{j}{\sum }C^{j}$$
(14)$$B=\frac{1}{\sum_{j}C^{j}}\underset{j\epsilon \left\{1,2,3\right\}}{\sum }\left(\gamma_{j}∙C^{j}∙B^{j}\right)$$
where C represents the fused score, B is the fused box parameter, and $\gamma_{j}$ is a parameter to adjust the different confidence weights. To be specific, two BRs are set on at the beginning $BR_{1}\;$and end of the refiner $BR_{2}$, respectivately. There $\gamma_{BR_{1}\;}=\left\{0.2,\;0.8\right\}$ $\gamma_{BR_{2}\;}=\left\{0.3,\;0.3,0.4\right\}$. And when $\gamma_{\;}=\left\{0.5,\;0.5\right\}$, Box Redistribution will transforme into box voting [17].

### 3.5. Loss Function

The network is trained end-to-end with a multi-task loss function. The loss function comprises two components, including object detection loss $l_{o}$ and grasp detection loss $l_{g}$. Since object detection refers to a multi-stage extraction ROI, the loss should be calculated for every ROI, such that the loss function for object detection is defined as:(15)$$l_{o}=\overset{3}{\underset{n=1}{\sum }}{\left(l_{cls}+\lambda l_{pos}\right)}_{n}$$ where $l_{cls}$ expresses the categorical cross-entropy loss of the object; $l_{pos}$ represents the Smooth-L1 loss between the predicted position and the ground-truth box; λ is adopted to adjust the weight occupied by the loss; n denotes the refiner number. The specific definition is illustrated as follows.
(16)$$l_{o}=\overset{3}{\underset{n=1}{\sum }}{\left(\frac{1}{N_{cls}}\underset{i}{\sum }l_{cls}\left(p_{i},p_{i}^{T}\right)+\lambda \frac{1}{N_{pos}}\underset{i}{\sum }p_{i}^{*}l_{pos}\left(t_{i},t_{i}^{T}\right)\right)}_{n}$$
where $N_{cls}$ and $N_{pos}\;$denote number of Classification and sampled region proposals on the training stage, repetitively. The Classification loss component $l_{cls}$ is written as follows:(17)$$l_{cls}\left(p_{i},p_{i}^{T}\right)=-\log\left[p_{i}^{T}p_{i}+\left(1-p_{i}^{T}\right)\left(1-p_{i}\right)\right]$$
where $p_{i}$ denotes the probability that the region is proposed to be predicted as the target region. The object localization loss$l_{pos}$ is written as:(18)$$l_{pos}\left(t_{i},t_{i}^{T}\right)=smooth\;L_{1}\left(t_{i}-t_{i}^{T}\right)$$
where $t_{i}$ denotes the four parameter coordinates $\left(x,\;y,\;w,\;h\right)$ of the rectangular box detected by the network; $\left(x,\;y\right)\;$represents the center point of the predicted object rectangular box;$\;\left(w,\;h\right)$ expresses the width and height of the measured object rectangular box; $t_{i}^{T}$ is the ground truth label.

For the grasping branch, the respective oriented anchor box ($p_{x},p_{y},p_{w},p_{h},p_{\theta }$) will correspond to a 5D offset $\left(\sigma_{x},\sigma_{y},\sigma_{w},\sigma_{h},\sigma_{\theta }\right)$ and a 2D grasping box confidence score vector ($\rho_{g},\rho_{ug}$). Thus, the grasp detection loss $l_{g}$ is defined as follows.
(19)$$l_{g}=l_{Gloc}+\alpha l_{Gcls}$$
(20)$$l_{Gloc}=\overset{P}{\underset{i\in Positive}{\sum }}\underset{m\in \left\{x,y,w,h,\theta \right\}}{\sum }smoothL1\left(\sigma m_{g}^{\left(i\right)}-\sigma m_{gt}^{\left(i\right)}\right)$$
(21)$$l_{Gcls}=-\overset{P}{\underset{i\in Positive}{\sum }}\log\left(\rho_{g}^{\left(i\right)}\right)-\overset{3P}{\underset{i\in Negative}{\sum }}\log\left(\rho_{ug}^{\left(i\right)}\right)$$
where $\sigma m_{gt}$,$\sigma m_{g}$∈{x,y,w,h,θ} express the grasping ground-truth offset and the predicted proposals offset, respectively. Every positive sample is set as an anchor matching at least one ground truth. An anchor is considered as a negative sample if not matching any ground truth value. Lastly, the top P positive samples and the top 3P negative samples are selected for classification training. Furthermore, the entire loss of the network is defined as follows.
(22)$$loss=l_{o}+\lambda l_{g}$$
where λ is all set to 1 in our experiments.

## 4. Experiment

In this section, we mainly discuss the details of our experiment. The multi-object dataset we use is VMRD. To verify the generalization capability of the model, experiments were also tested on our homemade test sets and the Kinova robot. The specific brief introduction to this section is presented as follows:

First, the dataset applied is presented in Section 4.1. Next, some details will be illustrated during the training stage in Section 4.2. Subsequently, the metric of validation is explained in Section 4.3. Afterward, we show the validation results on the VMRD in Section 4.4. Lastly, we make a series of ablation experiments to prove the effectiveness of our design in Section 4.5. Moreover, in order to ensure the practicality of our model, we have done a series of comparative experiments on the Kinova robot in Section 4.6.

### 4.1. Dataset

**VMRD dataset**. The v2 version of the multi-object VMRD [19,20,21] dataset comprises 4233 training data and 450 test data (RGB images). The dataset covers 31 object classes with 2–5 stacked objects in each image. The dataset is labeled with the true value of the grasping box and the corresponding category for the respective image. The "parent node" and "child node" rules are adopted to annotate each target pair. Actually, this rule specifies the order to follow when grasping different objects. Moreover, VMRD annotates both object kind and position. Since the original intention of the VMRD grasping dataset is to allow multi-objective grasping position detection in the position information of all objects. Furthermore, it is expected that the grasp detection algorithm is capable of acquiring the overall information in the picture rather than the local information.

### 4.2. Tarining Details

The two Nvidia 3090 graphics cards with 24 GB video memory were employed in this part of the experimental training, and the Pytorch network framework was adopted in the python2 environment, with stochastic gradient descent as the optimizer with an initial learning rate lr=0.0001 and a 10-fold decrease in the learning rate per 12 training rounds. The batch size was set to 3 for 36 training rounds. In the training process, NMS IoU 0.7 was set to keep 2000 proposals with a 1:1 ratio of positive and negative samples. SGD was employed as the optimizer, and the momentum was set to 0.9. Other settings are consistent with those of Cascade RCNN.

### 4.3. Evalution Metrics

The criteria for correct forecasting are specified as the following two requirements.

(1)The difference in angle between the predicted grasping proposals and the ground-truth box should be less than 30°.(2)The Jaccard intersection ratio between the predicted grasping proposals and the ground-truth box should not be less than 25%. The specific context is in Figure 4. A is the ground truth box, and B is the predicted grasping box.

The angle from the ground truth box is assumed to be $\theta_{1}$ and the angle from the prediction box to be $\theta_{2}$. On that basis, the difference between the two angles can be obtained as $\min\left(\left|\theta_{2}-\theta_{1}\right|,180^{{}^{\circ}}-\left|\theta_{2}-\theta_{1}\right|\right)$. Jaccard intersection ratio measures the correlation between two boxes.

Mean Average Precision (mAP) with grasp is employed as the evaluation metric for multi-target grasp position detection, which is expressed as follows:(23)$$MAP=\frac{\sum_{j\epsilon \left\{1,2\ldots .N_{m}\right\}}Ap^{j}}{N_{m}}$$ where $N_{m}$ denotes the total number of object types; $Ap^{j}$ expresses the Average Precisio from jth Category.

### 4.4. Evaluation on VMRD Dataset

This section aims to explore MMD in multi-object scences. Specifically, experiments will be performed on the multi-object grasping dataset VMRD to verify the effectiveness of the MMD proposed in this chapter.

The output of the actual prediction is expressed as follows:(24)$$f_{out}=\overset{n}{\underset{i=1}{\sum }}{\left(x,y,w,h,cls,x_{g},y_{g},w_{g},h_{g},\theta \right)}_{i}$$

The accuracy of grasping detection for various objects is obtained in Figure 5. We found that the shape of the object with the accuracy is random and does not have a special distribution rule, which indicates that the network has indeed learned the corresponding characteristics.

The advance of MMD is demonstrated, and our model was trained using all the training sets of the VMRD dataset. Figure 6 presents the sample test results of the model using the algorithm. The results of the VMRD testing set are listed in Table 1. our scheme achieves 76.71% mAPg on the VMRD grasping dataset.

### 4.5. Ablation Study

Experiments were performed on the VMRD dataset testing set.

(1)Effectiveness of FRM

We have done comparative experiments on FPM in different situations. The results are shown in Table 2. Although grasping detection results with FRM are better than without FRM, the object detection effect can be damaged. This result illustrates the effectiveness of our module design for grasp detection.

(2)Position of FRM.

We tested the performance of the model when the FRM was in different positions. The results are shown in Table 3. Our model performs best when FRM is applied to the refiner. Therefore, we adpot the FRM on refiner $S_{1}$ in this article. At the same time, we find that the choice to put FRM on refiner $S_{3}$ will seriously affect grasping performance. One possible explanation is that the feature map from the refiner $S_{3}$ is influenced by the residual convolution block $S_{5}.$

(3)Effectiveness of Box Redistribution

We also conducted an ablation study on the effectiveness of box redistribution, and the results are shown in Table 4. By adding this design, the accuracy of our model increases by 1.19 percentual points without FRM and by 0.69 percentual points with FRM, which indicates the effectiveness of the box redistribution.

(4)Parameters of Box Redistribution

A series of comparative experiments were set up to explore the effect of different parameters on network performance, and Table 5 lists the experimental results. In the proposed multi-stage network, the detection box from the later refiner has higher reliability. Moreover, the confidence level of the previous stage can avoid over-fitting. Given the result of the above analysis, the parameters in BR2 are empirically selected as 0.3, 0.3, 0.4, respectively. With this design, the proposed model achieves the optimal results, thus verifying the effectiveness of the parametric design.

(5)Environmental adaptability Research

Due to limited materials, two different testing sets were built in accordance with some of the testing images in VMRD. For the first testing set, 500 RGB images were selected randomly from the VMRD, and existing objects were adopted to mimic their placement and manner as much as possible. The only difference from the original data was that the background plate was changed (Figure 7). The second test set was built with a capacity of 200 using the same items as the VMRD free build, part of which is presented in Figure 8. In terms of the deep learning model, poor adaptability is a common and difficult problem to solve, thus revealing a significant degradation in the model’s performance when the environment changes slightly. Although our dataset is small, it is effective in detecting model environmental adaptation. It was confirmed that the proposed model is environmentally robust, and some grasps of that are illustrated as Figure 7 and Figure 8. A plausible explanation is that multi-stage selection and BR design increase system fault tolerance.

### 4.6. Experiment on Kinova Robotic Arm

This section consists of three parts. Our equipment condition is introduced in Section 4.6.1. Then, some details are explained for hand-eye calibration and camera calibration in Section 4.6.2 and Section 4.6.3. Finally, we show the experimental results and process of Kinova robot arm in Section 4.6.4.

#### 4.6.1. Details of Equipment

A 16.04 Ubuntu operating system was employed on the Kinova robot. The core processor of the robotic arm is an Intel i5-10400 CPU that made in China. Moreover, the Kinova robot is also equipped with a Kinova Mico2 six-degree-of-freedom robotic arm manufactured by kinavo company from Quebec, Canada, and an Intel Realsense D435 camera made in China.

#### 4.6.2. Hand-Eye Calibration

In this paper, the open source project named easy_hand-eye [1] is used to complete the hand-eye calibration, and the experiment adopts an automatic method without relying on additional hardware for eye-in-hand calibration. The camera is fixed at the end of the robot, and the coordinate system is shown in Figure 9.

We define the coordinate system of the calibration as K. As well as TNM is present the conversion matrix of N coordinate system to M coordinate system, where $M,N\in \left\{B,\;C,\;E,\;K\right\}$. During the calibration process, the position of the calibration board is kept fixed, and the camera pose is adjusted automatically or manually. Based on the statement above, the transformation of the calibration plate coordinate system relative to the base coordinate system of the manipulator can be defined as:(25)TKB=TEB 1∙TCE∙TKC 1=TEB 2∙TCE∙TKC 2

Then we invert both sides simultaneously,
(26)TEB 2 −1∙TEB 1∙TCE=TCE∙TKC 2∙TKC 1 −1

Let
(27)P=TEB 2 −1∙TEB 1
(28)Q=TKC 2∙TKC 1 −1
(29)X=TCE

Therefore
(30)PX=XQ where X is the parameters obtained through hand-eye calibration.

#### 4.6.3. Camera Calibration

The internal parameters of the camera are calibrated using the Zhang Zhengyou calibration method [23]. Firstly, Complete Realsense D435 installation driver under Ubuntu16.04. Then, it is necessary to start the camera and install Kalibr in the ROS environment. The calibration board is a black and white checkerboard with a size of 6 × 7 and a resolution of $0.029\times 0.029m^{2}$. The calibration data is recorded through the rosbag in the calibration tool. Then the calibration program completes the final calibration. Lastly, we get
(31)$$K=\left[\begin{array}{ccc} f_{x} & s & x_{0}\\ 0 & f_{y} & y_{0}\\ 0 & 0 & 1 \end{array}\right]=\left[\begin{array}{ccc} 624.1 & 0 & 318.54\\ 0 & 625.22 & 241.02\\ 0 & 0 & 1 \end{array}\right]$$

#### 4.6.4. Grasp Experiment on Knovia

In order to verify the utility of our model, we decided to do some experiments on stacked multi-object scenes. The experimental method is to select 2 to 3 kinds of objects and place them on the table, then randomly stack the objects. The robot needs to grasp sequentially according to the results of the grasping detection. The criterion for judging whether the grasping is successful or not is the same as above, and a total of 150 grasping experiments were carried out. Figure 10 depicts the actual scene multi-target grasp detection results. The grasping process shown in Figure 11.

During the experiment, the number of grasps and success for each object were counted. And the final results are shown in Table 6.

Experiments show that our approach is effective. In the stacking scene, it can complete the grasping task very well. This illustrates the practicality of our method.

## 5. Conclusions

In this study, a grasping detection algorithm based on multi-stage regress is proposed for the multi-object grasp position detection problem in complex environments. This algorithm is capable of detecting all object grasp positions and all object positions and types in the image. The algorithm is tested on VMRD, a multi object grab dataset. As indicated by the experimental results, the proposed algorithm outperforms existing multi-object grasp detection algorithms on the VMRD dataset. Moreover, to ensure the generalizability and practicality of the model, it was also tested on two test sets of our own making and on a kinova robot arm. The results reveal that MMD exhibits robust environmental adaptability and good practical performance. It also verifys the effectiveness of MMD. These good detection effects are closely related to FRM and BR modules. In fact, although our model has achieved a good grasping effect, it still faces the general problem of deep learning beacause the target category is fixed. At the same time, the impact of the selection of different backbone networks and network parameters has not been explored in this paper. In addition, our grasping order is still marked manually, which is also an urgent matter to be solved. Therefore, the follow-up research is still meaningful.

## Figures and Tables

**Figure 1 micromachines-14-00117-f001:**
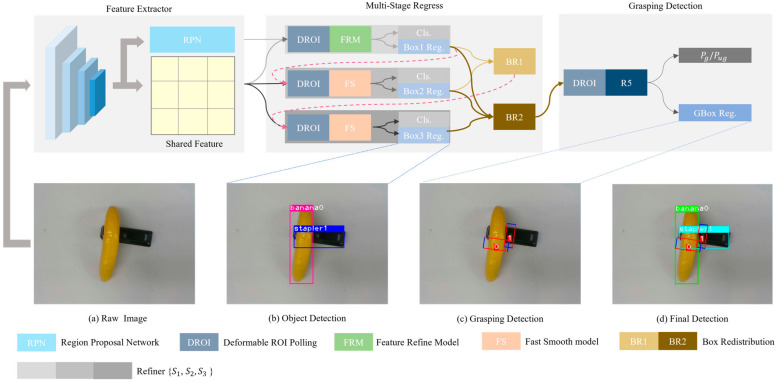
The overall architecture of the proposed MMD. The raw images (**a**) are first fed into 2D convolution network to generate Shared Feature and 2D object proposals. Subsequently, Shared Feature and 2D object proposals are passed into Multi-Stage Regress model to refine proposals Continuously and generate the predicted object Category and Bounding Box (**b**). Lastly, all object box features are aggregated by BR2 from every refiner to grasping detection to learn specific features for grasping box regress (**c**). The final detection result is illustrated as (**d**).

**Figure 2 micromachines-14-00117-f002:**
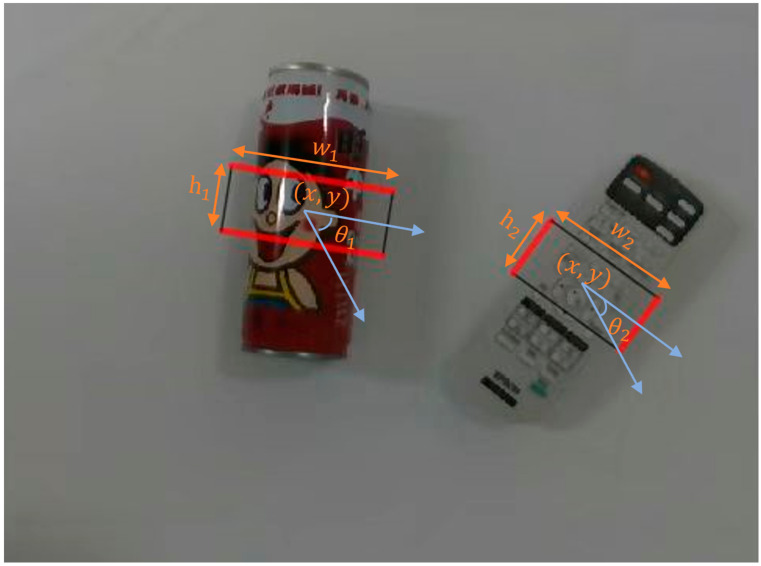
The representation of grasping box. Where $\left(x,\;y\right)$ is the box’s centre. W and h represent the length and width of the grasping box. θ is the angle formed by the grasping box and the horizontal axis of the camera.

**Figure 3 micromachines-14-00117-f003:**
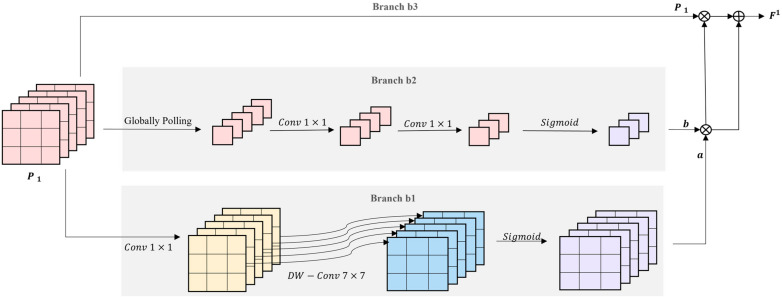
Feature refine model (FRM). The pooled features $P_{1}$ is fed into three branchs respectivately to aggregate a better region proposal feature map.

**Figure 4 micromachines-14-00117-f004:**
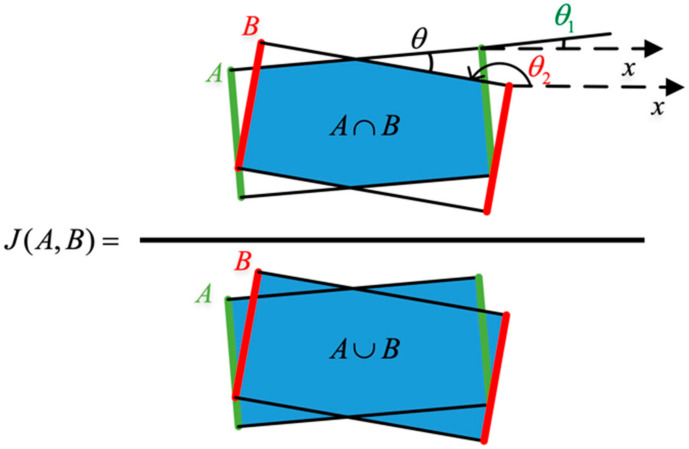
Jaccard coefficient.

**Figure 5 micromachines-14-00117-f005:**
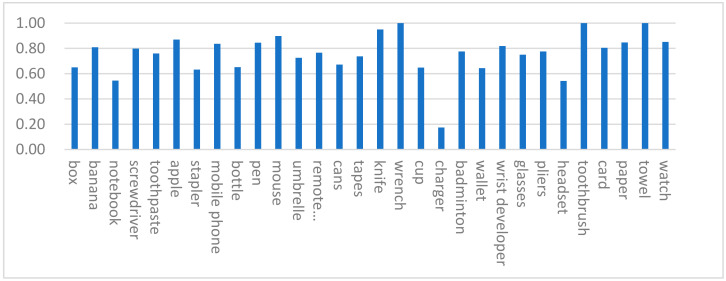
Object grasp position detection accuracy.

**Figure 6 micromachines-14-00117-f006:**
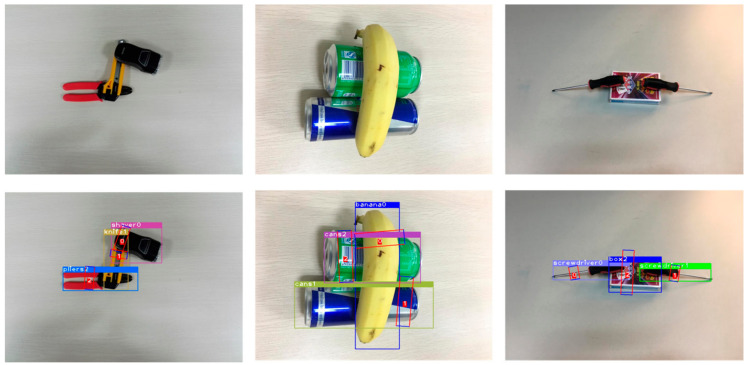
Sample test results of MMD. The three images above are the original images from the VRMD and the following are the results of the test.

**Figure 7 micromachines-14-00117-f007:**
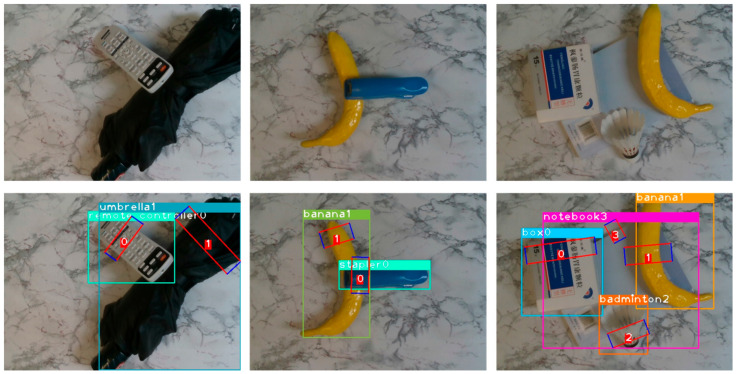
First test set sample. The three images above are the original images from the its and the following are the results of the test.

**Figure 8 micromachines-14-00117-f008:**
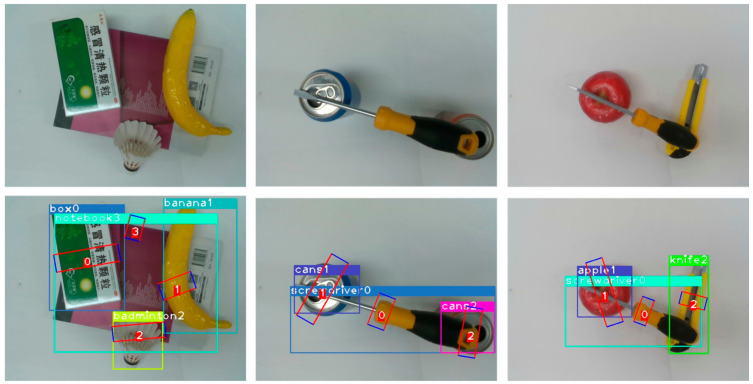
Second test set sample. The three images above are the original images from the its and the following are the results of the test.

**Figure 9 micromachines-14-00117-f009:**
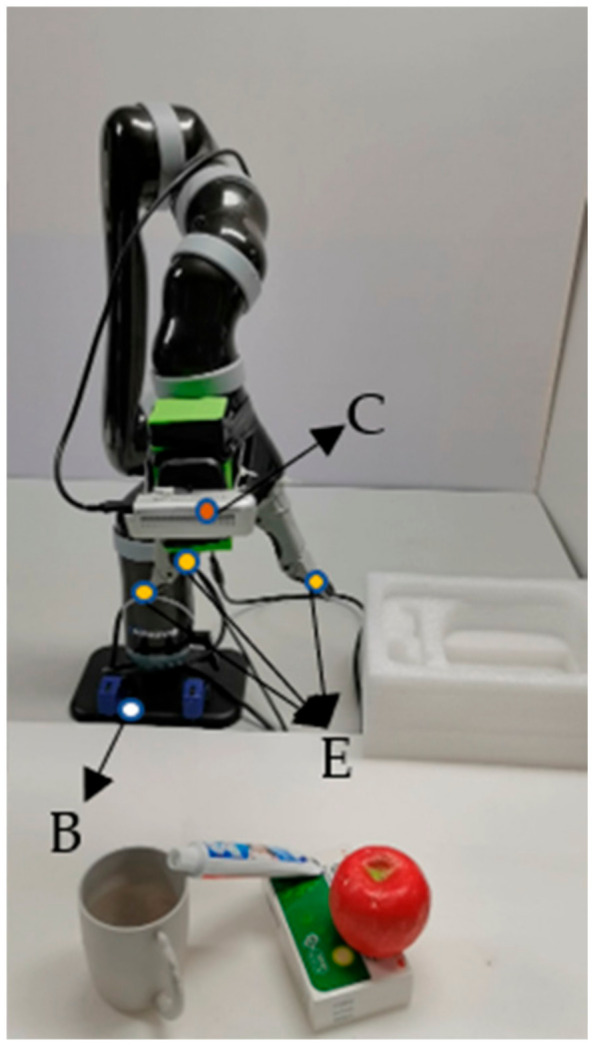
Coordinate System Declaration. Where B is the base coordinate system, E is the End Actuator Coordinate System, C is the camera coordinate system.

**Figure 10 micromachines-14-00117-f010:**
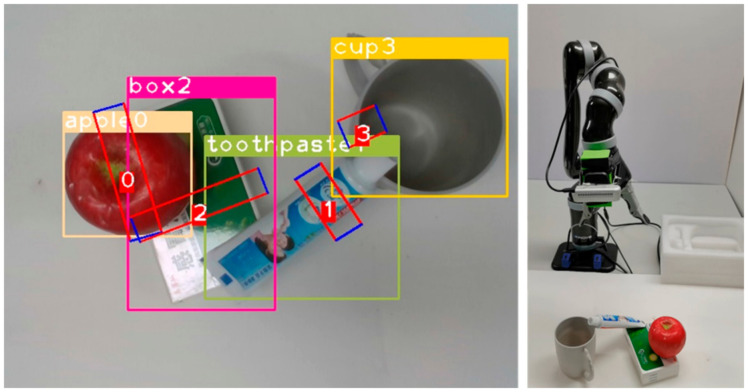
Multi-target grasp detection result.

**Figure 11 micromachines-14-00117-f011:**
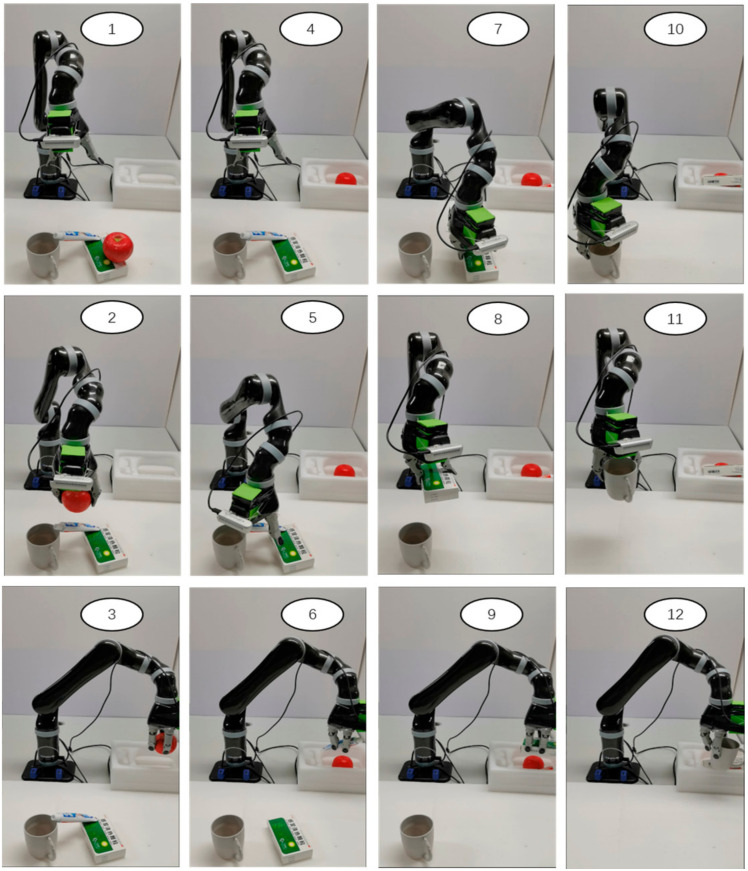
The grasp process in actual scences.

**Table 1 micromachines-14-00117-t001:** Performance of different algorithm on VMRD Dataset.

Algorithm	mAPg (%)
Faster-RCNN [9] + FCGN [18]	54.5
ROI-GD [22]	68.2
Zhang [5]	70.5
Keypoint-based scheme [7]	74.3
MMD	74.57
MMD + FRM	76.02
MMD + FRM + RR1 + RR2	76.71(+2.41%)

**Table 2 micromachines-14-00117-t002:** Effect of Feature Refine Model. Each colour represents a different comparison experiment.

FRM	RR1	RR2	mAPg (%)	mAPd (%)
			74.57	94.68
√			76.02	93.14
	√		75.73	94.46
√	√		76.68	92.56
		√	74.58	94.73
√		√	76.04	93.17
	√	√	75.76	94.48
√	√	√	76.71	92.86

**Table 3 micromachines-14-00117-t003:** Effect of Feature Refine Model at different stages.

$S_{1}$	$S_{2}$	$S_{3}$	mAPg (%)	mAPd (%)
			74.57	94.68
√			76.71	92.86
	√		75.10	92.91
		√	1.84	94.60
√	√		75.10	92.91
√		√	1.84	94.60
	√	√	1.69	92.88
√	√	√	1.69	92.88

**Table 4 micromachines-14-00117-t004:** Effect of Box Redistribution. Each colour represents a different comparison experiment.

FRM	RR1	RR2	mAPg (%)	mAPd (%)
			74.57	94.68
	√		75.73	94.46
		√	74.58	94.73
	√	√	75.76	94.48
√			76.02	93.14
√	√		76.68	92.56
√		√	76.04	96.17
√	√	√	76.71	92.86

**Table 5 micromachines-14-00117-t005:** Effect of different parameters on BR1.

$\gamma_{1}$	$\gamma_{2}$	mAPg (%)	mAPd (%)
0.1	0.9	75.61	93.13
0.2	0.8	76.71	92.86
0.3	0.7	74.35	92.63
0.4	0.6	74.03	92.51
0.5	0.5	72.12	91.9
0.6	0.4	72.05	91.91
0.7	0.3	72.94	91.7
0.8	0.2	72.25	91.2
0.9	0.1	73.29	91.15

**Table 6 micromachines-14-00117-t006:** Comparison of experimental results before and after improvement.

Method	Experiment Number	Number of Success	Grasp Success Rate (%)
Cascade R-CNN [10]	150	142	94.60
MMD + FRM + RR1 + RR2	150	146	97.33

## Data Availability

Not applicable.

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
