# Peer review of "A Practical Multi-Stage Grasp Detection Method for Kinova Robot in Stacked Environments"

_micromachines, 2022, doi:10.3390/mi14010117_

Round 1
Reviewer 1 Report
Review: A Practical Multi-Stage Grasp Detection Method for kinova robot in Stacked Environments
This paper proposes a multistage detection method based in a modified cascade Rcnn to determine the object and determine the orientation/location to better grasp by a robot manipulator. The pipeline is supported by a resnet101 to act as a feature extraction to be used by the pipeline to better determine the box and orientation of the object to grasp.
Concerning paper organisation, it follows a general organisation, introducing the problems and limitations slightly in applying the available conventional methods to the grasping task. The literature-related work is a bit incomplete.
Concerning the implemented pipeline, the multistage regression follows and explores the cons extracted features to generate the box and the class of the object. However, the BR2 features for grasping should be aligned with a defined objective. At least a short text clarification on some of the potential features useful to the grasping stage would clarify this stage and sustain its main objective. They called Box Redistribution to obtain more accurate regression. The image and text present an attention scheme to capture relevant information among the high and low-level features map.
In box redistribution, the problem of the box refiner is not clearly defined, the text is misleading. However, there is a missing problem that is not discussed in the entire paper, the order of picking/grasping. While the problem of predicting the best object location and orientation is discussed in the presented work, nothing is said about the grasping order. I understand that this problem is a complex problem by itself to be addressed in a single work, but a small mention to this problem would clarify the intention of the paper in only focusing in determining the best grasping location correctly and grasp order is not a focus in the presented work.
The results in figure 4 show that long narrow object locations are predicted with higher accuracy in comparison with others shapes. Perhaps the format of the object, long and sharp, allows a better definition of the angle of the object. This seems to be the confusion that the reader can reach, but it should be highlighted in the discussion of the results.
Results improvement should be presented in pp and not in percentages. When comparing the difference of two percentages, the results/discussion should be presented in percentual points (p.p) and not %.
While the proposed pipeline is coherent, however my concerns mainly include:
1. The modelling of the loss encompasses possible situations of symmetry? In narrow objects, there is a possibility of having two good optimal candidates por grasp. But the angle/orientation by itself only requires one definition (but the picking point can be two on same orientation).
2. In the experiments due to occlusions, there are cases of split detections, such as an object as a pen or pencil, due to occlusion being detected as two? This is a substantial problem when the environment has many overlapping objects. How was the model robust to this?
3. Conclusions should be elaborated more, with more pros and cons.
4. The order of picking is not discussed.
5. How the peorfmances is related to the backbone detector? This sure has some impact. DETR have been considered? there are a bit more robust.
Other Issues:
- In some figures references, there are ? When referencing the image in the text.
- In some parts of the text, the sentences are not clear, as an example:field, and the feature map is input into the…, location of objects in the image. R-CNN employs selective search to generate proposal regions, w The are many others like this
- English should be checked in the sentence construction description.
- Conclusions more elaborated and discuss of pros and cons with more substance.
Reviewer 2 Report
This practical development and application are valuable for researchers.
The paper has many mistakes, but the content is enough.
Please review the yellow highlights text.
For equation 22. Why is the value for Lamba = 1? What happens if the value is 0.5 or 1.5?

Round 2
Reviewer 1 Report
Reviewwers Assered the pending questions.
I would suggest to put the illustrative figure you send in the paper comments response of the kitchen fork in the paper and contextualized with some text. Is a nice clear clarification of the objective.
